# Barriers and facilitators to perioperative smoking cessation: A scoping review

**Sandra Ofori**[1]*, **Daniel Rayner**[2], **David Mikhail**[3], **Flavia K. Borges**[1], **Maura M. Marcucci**[1], **David Conen**[1], **Lawrence Mbuagbaw**[4], **P. J. Devereaux**[1]

1 Department of Medicine, McMaster University Canada, Hamilton, Canada, 2 Department of Health Research Methods, Evidence and Impact, McMaster University Canada, Hamilton, Canada, 3 Department of Health Sciences, McMaster University Canada, Hamilton, Canada, 4 Department of Health Research Methods, Evidence and Impact, McMaster University, Hamilton, Canada

* oforis@mcmaster.ca

## Abstract

### Objective

Smoking cessation interventions are underutilized in the surgical setting. We aimed to systematically identify the barriers and facilitators to smoking cessation in the surgical setting.

### Methods

Following the Joanna Briggs Institute (JBI) framework for scoping reviews, we searched 5 databases (MEDLINE, Embase, Cochrane CENTRAL, CINAHL, and PsycINFO) for quantitative or qualitative studies published in English (since 2000) evaluating barriers and facilitators to perioperative smoking cessation interventions. Data were analyzed using thematic analysis and mapped to the theoretical domains framework (TDF).

### Results

From 31 studies, we identified 23 unique barriers and 13 facilitators mapped to 11 of the 14 TDF domains. The barriers were within the domains of knowledge (e.g., inadequate knowledge of smoking cessation interventions) in 23 (74.2%) studies; environmental context and resources (e.g., lack of time to deliver smoking cessation interventions) in 19 (61.3%) studies; beliefs about capabilities (e.g., belief that patients are nervous about surgery/diagnosis) in 14 (45.2%) studies; and social/professional role and identity (e.g., surgeons do not believe it is their role to provide smoking cessation interventions) in 8 (25.8%) studies. Facilitators were mainly within the domains of environmental context and resources (e.g., provision of quit smoking advice as routine surgical care) in 15 (48.4%) studies, reinforcement (e.g., surgery itself as a motivator to kickstart quit attempts) in 8 (25.8%) studies, and skills (e.g., smoking cessation training and awareness of guidelines) in 5 (16.2%) studies.

### Conclusion

The identified barriers and facilitators are actionable targets for future studies aimed at translating evidence informed smoking cessation interventions into practice in perioperative

**Data Availability Statement:** All data files are available from the Open science Framework (OSF) database: https://osf.io/zpyd/?view_only= c621853d3da0494082da159907058be6.

**Funding:** The authors received no specific funding for this work.

**Competing interests:** The authors have declared that no competing interests exist.

settings. More research is needed to evaluate how targeting these barriers and facilitators will impact smoking outcomes.

## Background

Annually, over 300 million major surgical procedures are undertaken globally [1]. Surgery is an important point of contact between patients and the health care system and is an excellent opportunity to address the health needs of patients, as they are focused on their health during this time [2,3]. In North America, the prevalence of smoking among surgical patients is between 14% and 27% [4–6] compared to 13% to 14% of the general population [7]. Smoking is a leading cause of mortality and is associated with the risk of cardiac, infectious, and pulmonary complications, and delayed wound healing after surgery [8,9].

Smoking cessation involves the use of evidence-based pharmacotherapy and behavioral counseling [10–12]. Few surgical patients receive these interventions despite perioperative guidelines recommending them [13–15]. In an analysis from the VISION study, a large prospective cohort study of a representative sample of 40,004 patients aged ≥45 years who underwent major noncardiac surgery across 28 centers in 14 countries, we found that, among the 5480 (13.7%) who were smoking in the 4 weeks preceding their surgery, only 2.4% received any smoking cessation pharmacotherapies before surgery and 6.5% after surgery. Over 50% of the smokers resumed smoking after surgery, majority within 14 days [16].

Smoking cessation is complex and requires individual and system-led behavior changes [17,18]. Several studies have investigated possible barriers and facilitators in the implementation of perioperative smoking cessation interventions. Many of them from the perspective of specific subgroups of perioperative care providers like sub-specialty surgeons or nurses, limiting the interpretation of the results in a broader context [19–23].

The theoretical domains framework (TDF) is an evidence-based and comprehensive framework that can help understand the determinants of behavior change [24]. The TDF provides a structured approach to understanding the various factors that influence smoking behavior, including attitudes, beliefs, social influences, and environmental factors. The TDF offers a framework to develop more effective smoking cessation programs that consider these factors to improve cessation rates and reduce the harm caused by smoking [25]. Previous studies have utilized the TDF framework to identify key factors affecting smoking cessation in various groups. Campbell et al. found that existing smoking cessation methods for pregnant women don't sufficiently address social network influences, a major barrier [26]. Huddlestone et al. identified time constraints as the main obstacle for smoking cessation in mental health settings, with effective support materials being the most helpful enabler [27].

To our knowledge, no review has evaluated the barriers and facilitators of smoking cessation in the surgical setting. Therefore, we conducted scoping review of the existing literature to identify the barriers and facilitators to smoking cessation in the surgical setting, involving patients and the healthcare system and providers, and to organize this evidence into a theoretical-domains framework (TDF).

Review Objective: The objective of this scoping review was to summarize the quantitative and qualitative literature on the barriers and facilitators of smoking cessation among adults undergoing surgery.

## Methods

We conducted this scoping review according to the Joanna Briggs Institute (JBI) guidelines [28]. This method permits inclusion of all relevant literature and focuses on collating and summarizing the evidence. The manuscript was prepared according to the Preferred Reporting Items for Systematic Reviews and Meta-Analyses Extension for Scoping Reviews (PRISMA-ScR) guidelines (S1 Table) [29]. We did not register the protocol for this scoping review.

### Search strategy

We consulted a health sciences librarian with expertise in quantitative and qualitative methods, and developed a search strategy for MEDLINE, Embase, Cochrane Central Register of Controlled Trials (CENTRAL), CINAHL, and PsycINFO databases using keywords and MeSH terms (S2 Table).

The search was restricted to English language articles published in 2000 or later. We also manually-searched all reference lists of included studies to identify additional studies of relevance. We also screened systematic reviews for individual studies that met the eligibility criteria. The final searches were conducted on 11 August 2022.

### Eligibility criteria

We considered all published studies (quantitative and qualitative) involving adults (aged 18 or older) who smoke undergoing surgery or involving health care professionals who care for them. We included studies published in 2000 or later, in English, detailing barriers and facilitators of perioperative smoking cessation interventions from the perspective of health care providers or patients, with no geographic limitations. We included RCTs, prospective and retrospective cohort studies, cross-sectional studies, qualitative studies, and mixed-methods studies. We excluded studies not involving our population of interest, reviews, or editorials, and theses and conference abstracts that were not published in full.

### Study selection

Two reviewers independently conducted a pilot screening of 10 randomly chosen references to ensure consistent application of our eligibility criteria. Following a consensus meeting and final adjustments, the remaining titles and abstracts were screened independently by two reviewers on Covidence (Covidence systematic review software, Veritas Health Innovation, Melbourne, Australia. Available at www.covidence.org). Potentially relevant full text articles were independently assessed by two reviewers, and disagreements were resolved through discussion or by a third reviewer. We recorded the reasons for exclusion of full-text articles (S1 File).

### Data extraction

Two reviewers independently extracted data using a pre- piloted Excel form, including participant details, study methods, and key findings. The form was tested using two articles independently by two reviewers and revised in a meeting. Disagreements were resolved through discussion or third-party review.

### Appraisal of study reporting quality

Two reviewers independently, performed quality appraisal and reporting of methodological quality of the included studies using the Mixed Methods Appraisal Tool (MMAT). The MMAT facilitates the evaluation of the methodological quality of studies included in a review when it includes qualitative research, randomized controlled trials, non-randomized studies,

quantitative descriptive studies, and mixed methods studies [30]. There is no clearly defined scoring system with this tool, and we chose to report the final reporting quality as the percentage of the number of reporting items divided by the number of total items evaluated. We considered scores <60% as low, 60–80% as moderate or >80% as high quality. Disagreements were resolved by discussion or by a third reviewer.

## Data analysis and presentation

**Theoretical framework.** We used the TDF. This is a determinant framework that is used to describe factors (barriers or facilitators) that can influence behavior change. The TDF builds on 128 behavior change constructs found in 33 behavior change theories. These constructs are categorized into 14 theoretical domains; knowledge, skills, memory, attention and decision processes, behavioral regulation, social/professional role and identity, beliefs about capabilities, optimism, beliefs about consequences, intentions, goals, reinforcement, emotion, environmental context and resources, and social influences [24].

**Data synthesis.** Two reviewers conducted thematic analysis to identify common ideas and patterns from the articles. They grouped these into subthemes and broad themes, with disagreements resolved by discussion. We extracted data on what participants considered as barriers and facilitators to smoking cessation and we present the number of participants who express these within each study as numbers and percentages. We provide a descriptive summary of the number of identified articles that report barriers or facilitators from healthcare providers and patients' perspectives. We mapped all identified themes into the TDF domains and provide a descriptive summary highlighting the most frequent domains.

## Ethics and dissemination

Our study did not require ethics approval as we collected publicly available data.

## Results

### Description of studies

Our database search yielded 6,962 publications. We screened 5,771 titles and abstracts after we removed duplicates and included 122 articles for full text review. Thirty-one articles met the eligibility criteria to be included in this review (Fig 1).

The 31 studies were published between 2000 and 2022. There were 20 cross-sectional surveys, 2 randomized controlled trials (RCTs) that included a qualitative sub-analysis of barriers and facilitators, 4 qualitative and 5 mixed methods studies. Of the 31 studies, 12 included patients [31–42], 6 included surgeons [43–48], 5 included nurses [44–53], 2 included anesthesia providers [22,54], 5 included a mix of health care providers (HCPs) including primary care physicians, nurses, anesthesiologists, and surgeons [19,23,55–57], and 1 recruited surgical residency program directors [58]. Overall, 11,871 participants (10,407 HCPs, and 1,464 patients) were recruited across the 31 included studies (Table 1). Of all the included studies, only one was theory based. The authors cited that they were guided by "social cognitive theory and the conflict theory of decision making" [41].

### Reporting quality

Of the 31 articles, 10 (32.3%) articles scored >80% for reporting quality and 21 (67.7%) scored 60–80%. None scored <60%. The average reporting quality of the 33 articles was a mean of 85% ± 11% (S3 Table).

**Barriers and facilitators to perioperative smoking cessation: a scoping review**

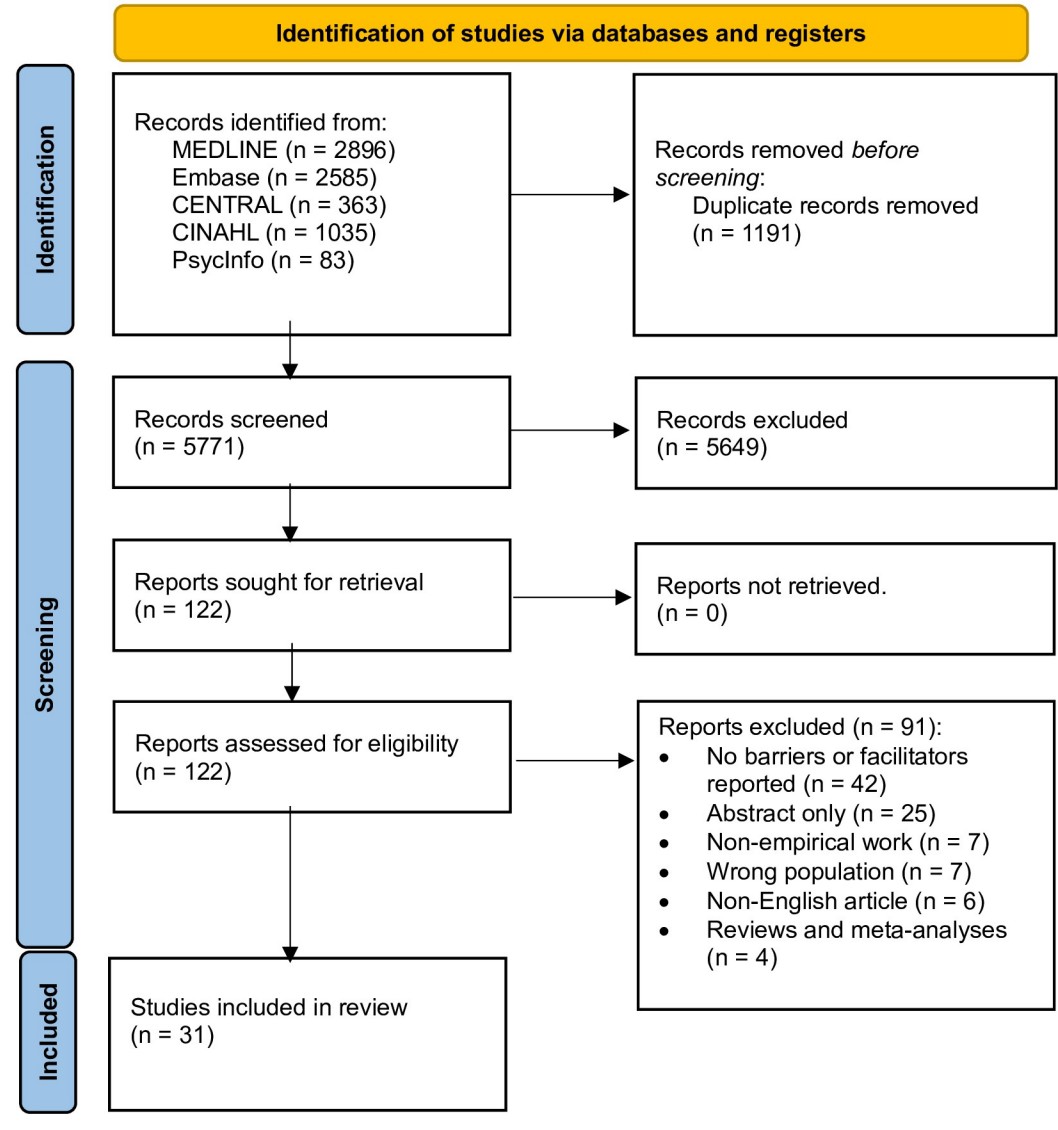

**Fig 1. PRISMA flowchart.**

### Identified barriers and facilitators

Our review identified 23 barriers and 13 facilitators to perioperative smoking cessation mapped to 11 of the 14 TDF domains (Table 2 and Fig 2). The most frequent domains for identified barriers and facilitators were the environmental context and resources (31 studies, 3 barriers, 7 facilitators), knowledge (23 studies, 4 barriers, no facilitators), beliefs about capabilities (15 studies, 3 barriers, 1 facilitator), and social/professional role and identity (9 studies, 2 barriers, 1 facilitator) domains (S1 Fig).

**Table 1. Study characteristics.**

| Author/Year | Country | Study design | Participants | Sample size |
|---|---|---|---|---|
| Crews 2008 [43] | USA | Cross-sectional survey | Oral and maxillofacial surgeons | 2740 |
| Taniguchi 2011 [50] | Japan | Cross-sectional survey | Nurses | 2215 |
| Yao 2009 [48] | China | Cross-sectional survey | Male surgeons | 823 |
| Schultz 2014 [58] | USA | Cross-sectional survey | Anaesthesiology residency program directors and residents | 528 |
| Smeds 2017 [34] | USA | Cross-sectional survey | Vascular surgery patients | 490 |
| Shi 2010 [3] | China | Cross-sectional survey | Anaesthesiologists | 482 |
| Houghton 2008 [49] | USA | Cross-sectional survey | Certified registered nurse anaesthetists | 439 |
| Yankie 2006 [51] | USA | Cross-sectional survey | Certified registered nurse anaesthetists | 271 |
| Yu 2013 [37] | China | Cross-sectional survey | Adults scheduled for elective non-cardiovascular surgery | 227 |
| Gay-Escoda 2012 [44] | Spain | Cross-sectional survey | Oral surgeons | 224 |
| Marrufo 2019 [45] | USA | Cross-sectional survey | Thoracic surgeons | 200 |
| Webb 2013 [36] | Australia | Cross-sectional survey | Current smokers and those who reported quitting smoking for surgery planning to undergo surgery | 177 |
| Newhall 2017 [31] | USA | Pilot, multi-center, cluster RCT | Adult smokers with PAD | 156 |
| Bottorff 2016 [40] | Canada | Cross-sectional surveys and interviews | Smokers undergoing elective surgery | 150 |
| Saddichha 2010 [47] | India | Cross-sectional survey | Dental surgeons | 100 |
| Owen 2007 [46] | UK | Cross-sectional survey | Non-vascular surgeons | 83 |
| Shannon-Cain 2002 [33] | USA | Cross-sectional survey | Smokers undergoing any elective outpatient surgical procedure | 81 |
| Rajae 2019 [32] | USA | Parallel group RCT | Patients with peripheral arterial and aneurysmal disease | 59 |
| Van Slyke 2017 [35] | Canada | Cross-sectional cohort | Cosmetic surgery patients | 47 |
| Rosvall 2017 [53] | Sweden | Mixed methods questionnaires and semi-structured interviews | Surgical nurses | 47 |
| Jose 2020 [52] | USA | Mixed methods Focus groups | Perioperative registered nurses | 39 |
| Farley 2016 [38] | UK | Qualitative semi-structured interview | Surgical lung cancer patients | 22 |
| McDonnell 2014 [41] | USA | Mixed methods prospective, one-group repeated measures design with questionnaires and exit interviews | Patients scheduled for surgery for a suspicious thoracic mass or known cancer and family members | 16 |
| McDonnell 2016 [42] | USA | Mixed methods prospective, one-group repeated measures design with questionnaires and exit interviews | Patients scheduled for surgery for a suspicious thoracic mass or known cancer and family members | 16 |
| Kai 2008 [22] | Japan | Cross-sectional survey | Anaesthesiologists and thoracic surgeons | 1083 (542 anaesthetists; 541 surgeons) |
| Warner 2004 [20] | USA | Cross-sectional survey | Anaesthesiologists and general surgeons | 335 Anaesthesiologists, 359 general surgeons |

*(Continued)*

**Table 1.** (Continued)

| Author/Year | Country | Study design | Participants | Sample size |
|---|---|---|---|---|
| Karabeyoglu 2014 [55] | Turkey | Cross-sectional survey | Anaesthesiologists and general surgeons | 274 (108 anaesthesiologists, 93 general surgeons, 41 anaesthesiology residents, 32 general surgery residents) |
| Vick 2011 [19] | USA | Cross-sectional survey | Surgeons and anaesthesia providers, including both anaesthesiologists and nurse anaesthetists | 92 (55 surgeons, 22 anaesthesia providers, 15 primary care providers) |
| Luxton 2019 [23] | UK | Qualitative One-on-one, semi-structured interviews | Cardiothoracic surgeons, anaesthetists, nurses, and physiotherapists | 52 (15 cardiothoracic surgeons, 15 consultant anaesthetists, 11 nurses, 11 physiotherapists) |
| Warner 2008 [39] | USA | Qualitative semi-structured interview | Smoking surgical patients, staff anaesthesiologists and surgeons | 29 (19 cigarette smokers, 5 anaesthesiologists, 5 surgeons) |
| Newhall 2016 [57] | USA | Qualitative Focus groups | Vascular Quality Initiative representative, Tobacco cessation counsellor, Tobacco Quit Line representative, Vascular surgeons, Vascular surgery patients | 15 (2 tobacco cessation counsellors, 1 Quit Line representative, 1 Vascular Quality Initiative leader, 7 vascular surgeons, 4 patients) |

*UK- United Kingdom; USA- United States of America; RCT- Randomized controlled trial.*

### Environmental context and resources

The most frequent barriers and facilitators identified in this review were within the environmental context and resources domain, which refers to circumstances or contexts that influence the development of skills. "Time constraints" was the most reported theme across 11 studies, with 12%-48% of anesthesiologists, 7%-49.5% of surgeons, and 50–76% of nurses citing lack of time as a significant barrier to smoking cessation interventions [22,23,43,52]. Patients having surgery often have comorbidities that are prioritized over smoking cessation for example, in one study, a surgeon reports *"the discussion of stopping smoking has to be made but invariably the surgeons don't have time to do it. I have a 45-minute consult and invariably I run over. There's a lot to talk about in the management of their disease"*[23].

The theme "availability of smoking cessation support services" was reported as a barrier to effective smoking cessation by surgeons, anesthesiologists, and residency program directors. Lack of hospital support regarding accessibility and availability of quit kits, Quitline referral materials, nicotine replacement therapy, and lack of integration of care between all professionals and hospital environments involved in the patient's care were some of the identified barriers within this theme [23]. In one study, vascular surgeons noted it was often unclear what happened to their Quit Line referrals [57]. Clinicians reported medication cost as a barrier, but patients did not mention it in any of the reviewed studies. Clinicians suggested that more patients could be helped if cost was not an issue [57]. In one study of 2,740 surgeons in the United States of America (USA), 40.2% strongly agreed with the statement "*reimbursement issues prevent me from providing tobacco-use cessation services*" [43].

The theme "person and environment interaction" was reported by clinicians and patients to be a facilitator. The dominant subtheme, mostly reported by patients, was physicians (especially surgeons) proactively providing advice to quit smoking in a sensitive manner and offering help (e.g., NRT) to quit [33,38,40,49,57]. This is exemplified by a quote from a post-surgical patient, *"surgeons are probably the most influential people. . . . [I have a] great respect for their ability and what they did. That is part of selling something like this; we naturally put a lot of stock in the credibility of the person telling us"* [59]. In a Canadian study of a "stop smoking before surgery" program, patients were asked to provide feedback to improve the program and most recommended that physicians and surgeons should inform patients about the benefits of quitting smoking [40]. Multiple media sources, including mail, were suggested to

**Table 2.  Themes and sub-themes inductively generated for each TDF domain.**

| DOMAIN & THEMES | SUB-THEME BARRIER/FACILITATOR | PERSPECTIVE N = number of studies | | | | | | |
|---|---|---|---|---|---|---|---|---|
| | | Surgeons | Anaesthesia providers | Nurses | Mixed healthcare provider | Patients | Other | Total |
| **1. Environmental context and resources (n = 31 studies)** | | | | | | | | |
| Time constraints | *Barrier*: lack of time, limited time for face-to-face visits and counselling | 3 | 2 | 2 | 3 | 0 | 1 | 11 |
| Availability of smoking cessation support services | *Barrier*: lack of smoking cessation program in the hospital | 1 | 0 | 0 | 1 | 0 | 1 | 3 |
| Cost constraints | *Barrier*: cost of medications, lack of reimbursement for smoking cessation service provided | 1 | 0 | 0 | 1 | 0 | 0 | 2 |
| Integration of smoking cessation into perioperative workflow | *Facilitator*: seamless integration to link patients with external resources | 0 | 0 | 0 | 1 | 1 | 0 | 2 |
| Person/environment interaction | *Facilitator*: Sensitive and proactive approach to smoking cessation by healthcare provider | 0 | 0 | 0 | 2 | 1 | 0 | 3 |
| | *Facilitator*: concurrent receipt of smoking cessation advise from friends and family | 0 | 0 | 0 | 0 | 1 | 0 | 1 |
| | *Facilitator*: providers' positive attitude towards smoking cessation counselling | 0 | 0 | 1 | 0 | 0 | 0 | 1 |
| | *Facilitator*: physicians and/or surgeons should provide advice to quit smoking as part of routine care | 0 | 0 | 0 | 1 | 4 | 0 | 5 |
| | *Facilitator*: utilization of multiple media sources to provide smoking cessation advise | 0 | 0 | 0 | 1 | 1 | 0 | 2 |
| | *Facilitator*: individualization and patient specific timing of the approach to smoking cessation | 0 | 0 | 0 | 1 | 0 | 0 | 1 |
| **2. Knowledge (n = 23 studies)** | | | | | | | | |
| Knowledge of benefits | *Barrier*: perceived health benefits of smoking | 0 | 0 | 0 | 0 | 2 | 0 | 2 |
| Lack of adequate knowledge of smoking cessation support | *Barrier*: knowledge of available smoking cessation interventions | 2 | 2 | 1 | 4 | 1 | 0 | 10 |
| | *Barrier*: limited knowledge of smoking cessation counselling | 0 | 2 | 0 | 3 | 0 | 0 | 5 |
| | *Barrier*: perceived lack of efficacy of smoking cessation interventions | 2 | 2 | 1 | 1 | 0 | 0 | 6 |
| **3. Beliefs about capabilities (n = 15 studies)** | | | | | | | | |
| Perceptions about patient's capacity to quit | *Barrier*: patients are already nervous or upset about surgery/diagnosis | 1 | 2 | 2 | 1 | 1 | 0 | 7 |
| | *Barrier*: patients lack willpower | 1 | 0 | 0 | 0 | 1 | 0 | 2 |
| Perceptions about providers capacity to deliver smoking cessation support | *Barrier*: lack of providers' self-efficacy to provide treatment | 1 | 0 | 2 | 0 | 1 | 1 | 5 |
| | *Facilitator*: Physicians' belief that their advice is useful | 0 | 0 | 0 | 1 | 0 | 0 | 1 |
| **4. Social/professional role and identity (n = 9 studies)** | | | | | | | | |
| Perception of self | *Barrier*: role: providers not believing it is their role to provide smoking cessation treatments | 2 | 0 | 2 | 1 | 0 | 1 | 6 |
| Role of institution in addressing tobacco use | *Barrier*: providers not delivering advice about smoking cessation benefits | 0 | 0 | 0 | 0 | 2 | 0 | 2 |
| | *Facilitator*: established collaborative smoking cessation program | 0 | 1 | 0 | 0 | 0 | 0 | 1 |
| **5. Reinforcement (n = 8 studies)** | | | | | | | | |

(*Continued*)

**Table 2.** (Continued)

| DOMAIN & THEMES | SUB-THEME BARRIER/FACILITATOR | PERSPECTIVE N = number of studies | | | | | | |
|---|---|---|---|---|---|---|---|---|
| | | Surgeons | Anaesthesia providers | Nurses | Mixed healthcare provider | Patients | Other | Total |
| Incentives for quitting smoking | *Facilitator*: surgery itself as a motivator to kickstart quit attempts | 0 | 1 | 0 | 3 | 2 | 1 | 7 |
| | *Facilitator*: counselling/advice on surgery-specific benefits of smoking cessation | 0 | 0 | 0 | 0 | 1 | 0 | 1 |
| **6. Beliefs about consequences (n = 7 studies)** | | | | | | | | |
| Belief about usefulness of smoking cessation delivery before surgery | *Barrier*: doubts about benefits of short-term abstinence | 0 | 1 | 1 | 0 | 0 | 0 | 2 |
| | *Barrier*: smoking cessation interventions not worthwhile given short time before surgery | 0 | 0 | 1 | 0 | 0 | 0 | 1 |
| | *Barrier*: underestimation of the effect of smoking cessation on postop complications | 1 | 0 | 0 | 0 | 0 | 0 | 1 |
| | *Barrier*: indication for surgery not related to smoking | 0 | 0 | 1 | 1 | 0 | 0 | 2 |
| Anticipated regret | *Barrier*: discomfort disclosing smoking status | 0 | 0 | 0 | 0 | 1 | 0 | 1 |
| **7. Skills (n = 5 studies)** | | | | | | | | |
| Training and competency to deliver smoking cessation interventions | *Facilitator*: smoking cessation training and awareness of smoking cessation guidelines | 1 | 1 | 2 | 0 | 0 | 1 | 5 |
| **8. Intentions (n = 4 studies)** | | | | | | | | |
| Lack of intention | *Barrier*: providers' belief they cannot influence patients' decision | 1 | 0 | 0 | 0 | 0 | 0 | 1 |
| Perceived readiness to change | *Barrier*: perception that patients don't want to quit | 1 | 1 | 0 | 1 | 0 | 0 | 3 |
| **9. Social influences (n = 4 studies)** | | | | | | | | |
| Normative behaviour | *Barrier*: smoking behaviour of the healthcare provider | 2 | 1 | 0 | 0 | 0 | 0 | 3 |
| Smoking among social networks | *Facilitator*: smoking cessation interventions that involve family members who smoke | 0 | 0 | 0 | 0 | 1 | 0 | 1 |
| **10. Optimism (n = 3 studies)** | | | | | | | | |
| Pessimism interfering with participation | *Barrier*: Fatalistic given diagnosis requiring surgery | 0 | 0 | 1 | 1 | 1 | 0 | 3 |
| **11. Emotion (n = 2 studies)** | | | | | | | | |
| Coping mechanisms for stress | *Barrier*: smoking to cope with psychological stress of illness | 0 | 0 | 0 | 0 | 1 | 0 | 1 |
| Lack of meaningful activities | *Barrier*: smoking as only source of enjoyment | 0 | 0 | 0 | 0 | 1 | 0 | 1 |
| **12.** Goals (n = 0 studies) | | | | | | | | |
| **13.** Memory, attention, and decision processes (n = 0 studies) | | | | | | | | |
| **14.** Behavioural regulation (n = 0 studies) | | | | | | | | |

*other: Includes residency program directors.

disseminate advice to patients due to the limitations of the perioperative environment. Individualization of the timing of smoking cessation advice/interventions was also identified as a facilitator.

**Knowledge.** Several studies reported the following barriers to smoking cessation delivery in the perioperative setting: inadequate knowledge of the five steps for smoking intervention known as the 5A's (Ask, Advise, Assess, Assist, and Arrange treatment) and other counseling

**Barriers and facilitators to perioperative smoking cessation: a scoping review**

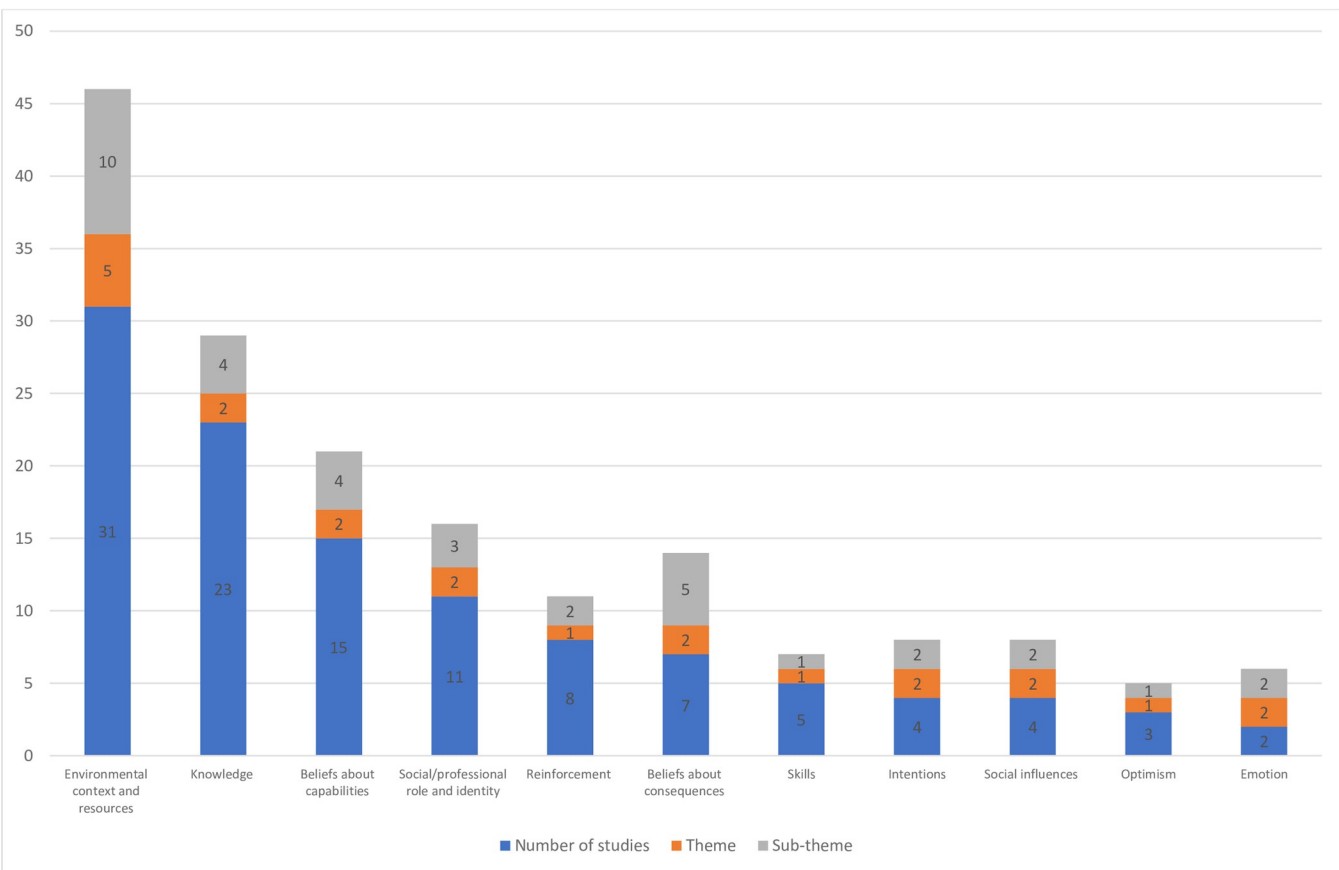

**Fig 2. Frequency of identified themes and sub-themes coded to TDF domains.**

techniques [19,20,22,23,39], and limited knowledge of NRT efficacy and how to help smokers [20,55]. Providers also believed that perioperative smoking cessation interventions were not efficacious [20,22,33,43,46,49,54]. For example, in a survey of 2,740 oral and maxillofacial surgeons in the USA, only 18% agreed that that counseling to stop smoking was "very or quite effective," and only 10.6% agreed that counseling patients to remain abstinent was "very or quite effective" [43]. In another study of 83 non-vascular surgeons in the United Kingdom, 45% did not know that counselling patients increased the likelihood of quitting and 48% did not know that augmenting advise with the provision of NRT and referral to cessation services increases the likelihood of successful cessation attempts [46]. Patients had a misconception that smoking was beneficial for their health around the time of surgery, e.g., one patient said, *"smoking makes him cough and clears his lungs of mucus"* and another reported *"quitting smoking will cause me to gain weight and worsen my mobility issues"* [38].

**Beliefs about capabilities.** Perceptions about patient's capacity to quit was identified as a barrier to abstinence in 15 studies. This included beliefs that patients lack willpower [43], as well as providers reporting that patients are stressed and anxious about surgery hence smoking cessation should not be discussed [22,43,49,52,54,55]. Furthermore, across the studies, the providers' lack of belief in their self-efficacy to provide smoking cessation treatment was a barrier. For example, only 30% of 443 nurses [49], 39% of 490 anesthesiologists [58] and 20% of 2,740

surgeons [43] believed they had adequate self-efficacy to provide smoking cessation treatments to patients who smoke.

Conversely, physicians are more likely to provide smoking cessation advice to patients when they believe it is useful, which is a facilitator. In a study involving 92 perioperative healthcare providers, those who believed that counseling patients would increase quit rates were significantly more likely to advise patients to quit, with 52.1% of believers taking time to counsel their patients compared to 0.0% of non-believers [19].

**Social/Professional role and identity.** Providers' perception that it is not their role to provide smoking cessation interventions was identified as a barrier in six studies. Clinicians sometimes lacked clarity on which professionals had the responsibility to provide support like pharmacotherapy, beyond giving advice. For example, one study [23] reported these quotes from a nurse: *'I think clinicians have a standard by-line 'You should quit smoking as it is bad for you'. In terms of committing to other therapies to help them or directing them to what will help them quit, it's unclear who does that"* and from a surgeon *"there's patches and gum and electronic cigarettes and cold turkey or whatever. I've got no idea, and I certainly don't prescribe it. So, I advise them to talk to their GP."* Across some of the studies, there was a low proportion of anesthesiologists (42% of 490) [58] and nurses (22% of 443) [49] who felt it was their responsibility to help patients get help. The proportions of surgeons who felt it was their responsibility was slightly higher, 59.4% of 2740 oral and maxillofacial surgeons [43] and 76% of 224 oral surgeons [44] in the two studies involving surgeons that identified this barrier. Conversely in another study, 59.8% of 276 nurses believed it was their responsibility and this was associated with a statistically significant increase in the likelihood of providing smoking cessation counselling [51].

Patients perceive it is the role of clinicians to provide smoking cessation advice and when this is not done, patients identify this as a barrier [38]. In one study, 46% of 150 patients did not receive advise from the clinician and were surprised that this was not discussed [40]. For example, one patient who independently learnt of the benefits of smoking cessation reported that *"Yeah, I was kind of surprised as I figured that people would be, you know, really all over the fact that I was a smoker and stuff, but no one really said anything."*

From an institutional perspective, the existence of a multi-disciplinary team of surgeons, anesthetists and specialist nurses facilitate the cohesive delivery of the same smoking cessation interventions and avoids mixed messages [23]. A focus group study involving vascular surgeons identified that they believed their responsibility for facilitating smoking cessation treatment should be limited to initiating the conversation with patients and obtaining their agreement to quit. The patients should then be referred to another healthcare provider like a tobacco specialist nurse to ensure that they receive the counselling and treatments. One participant noted *"if it's taken out of the surgeons' hands, it's the best thing,"* . . . . . . *"They should be able to say, 'You need to quit smoking, and here's a nurse who can help"* [57].

**Reinforcement.** Only facilitators were identified in the domain of reinforcement, from the perspectives of both patients and providers. The theme "incentives for quitting smoking" had two subthemes. Firstly, patients viewed surgery as a major life event, and receiving information about the surgery-specific benefits of quitting smoking served as a motivator to quit [41,57]. In one qualitative study of patients undergoing elective surgery at two hospitals in British Columbia, Canada, most patients agreed that surgery provided new or additional motivation for them to reduce or quit smoking and just learning about the benefits of cessation before surgery prompted an immediate quit attempt [40]. In another survey of 59 patients with peripheral vascular disease having open or endovascular procedures, 77% reported that they reduced smoking because of the surgical intervention itself [32]. Secondly, providers

believed the preoperative period is an opportune time to use impending surgery as an incentive for patients to quit smoking permanently [20,22,58].

**Belief about consequences.** Only barriers were identified within this domain, across 7 studies. The main theme was "belief about usefulness of smoking cessation delivery before surgery". Some providers expressed doubts about the benefits of short-term abstinence around the time of surgery, and the efforts to initiate smoking cessation were not worthwhile given the short time patients are seen before surgery [51,54]. Also providers underestimate the effect of smoking cessation on postoperative complications [46]. Sometimes, smoking cessation was not discussed as providers felt the indication for surgery was not related to smoking and that the health issue requiring surgery took precedence [19,50].

The theme "anticipated regret" within this domain shows that patients may feel regret about their smoking and disclosing this to their healthcare provider. This feeling of discomfort could be a barrier, stemming from a sense of self-failure. In a qualitative study of 22 lung cancer patients, a patient reported *"I felt ashamed. I've done this to myself you know. It's my fault I've got it. . .I don't think anyone realizes how addictive it is, and I feel it should be banned, cause it's a drug, you know. . . I couldn't give it up so. But I did in the end, but too late"*. It could also stem from feelings of guilt or fear of disappointing their health care provider for example, another patient in the study reported *"It's hard because [my GP has] been very good with me. You feel like you're wasting their time, you feel guilty, but she was great with me and she. . .asked me what I wanted to help me stop"* [38].

**Skills.** Across 5 studies, surgeons, anesthesiologists, and nurses identified that training on smoking cessation [43,51,54,58] and awareness of smoking cessation guidelines [22,43,51,58] were important facilitators. In two studies, this was associated with increased provision of smoking cessation counselling, fewer perceived barriers, and an increase in self-efficacy to deliver smoking cessation interventions among nurses and surgeons [43,51].

**Intentions.** Providers' beliefs about their inability to influence patients and their perceived patients' lack of readiness to quit, were barriers in this domain. In one study of 200 thoracic surgeons, 160 (80%) believed that patient's unwillingness to quit smoking was the main barrier interfering with their ability to help them quit smoking [45]. Another study showed that only 27% of 823 surgeons always advised patients to quit smoking. Surgeons who didn't advise patients to quit smoking, gave not being able to influence them as the main reason (39.4%) [48].

**Social influences.** The theme "normative behavior" refers to how the smoking norms, attitudes, and behaviors of healthcare providers impact their willingness to provide smoking cessation care. Providers who use tobacco are less likely to believe in the harms of smoking and less likely to provide cessation interventions [22,43,48]. In a survey of 542 anesthesiologists and 521 surgeons, physicians who smoked were less likely to agree that the perioperative period was a good time to help patients stop smoking (15% of smokers vs 36% and of non-smokers, P<0.0001), less likely to agree it was their responsibility to advise patients to quit (15% versus 27%, P = 0.01), and less likely to oppose strict hospital smoking policies (23% vs 52%, P<0.0001) [22].

The involvement of a patient's social network to provide social support was a facilitator from patients' perspective. In a study involving 8 patients and their family members who smoked in a thoracic surgery clinic, a patient reported, *"I was confident I could stop smoking this time, knowing that we were both going to quit together. We very seldom do anything without each other."* His spouse stated, *"At first, I was very skeptical. I thought I couldn't do it. But overcoming this together gave us the needed support. I'm confident now that we will stay smoke-free"* [41].

**Emotion.** The two themes in this domain came from one study involving surgical lung cancer patients and were barriers to smoking cessation intervention in the perioperative

period [38]. Patients identified that smoking was used as a mechanism to relieve the stress of their illness/diagnosis and a way to retain some enjoyment in their lives. For example, one participant in the study reported, *"I'm 66. . .I don't expect to live much longer. And when you see this world, I don't know whether I want to. I'm alright. . . . I just want to be happy and if a cigarette makes me happy, why shouldn't I have one. . . .I know I'm naughty and I shouldn't do it. . . but I'm ok"*.

## Discussion

What is new: our study is the first theory-informed scoping review examining barriers and facilitators to smoking cessation in the perioperative setting. Key finding include: i.) more barriers (23) than facilitators (13) to perioperative smoking cessation were identified; ii.) primary barriers encompass lack of time (number of studies n = 13), limited knowledge among providers (n = 10), and belief about patient apprehension about pre-surgery cessation discussions (n = 7); iii.) significant facilitators are using surgery as a motivation to quit (n = 7) and physician-provided cessation advice(n = 5). Notably, the link between these factors and cessation success remains unexplored. Our findings indicate a need for a comprehensive approach to implement effective perioperative smoking cessation strategies, addressing multiple identified barriers across several domains.

Lack of time and availability of smoking cessation support services were important barriers. System-level changes such as the establishment of smoking cessation programs and increasing clinic time for providers are needed, along with reimbursement for counselling patients or innovative solutions like the use of computer-based programs [60] or virtual care models [61] to deliver smoking cessation to patients. Except for one study [47], all were conducted in high and upper-middle-income countries. Since reimbursement systems differ across healthcare settings, any system-level changes must be tailored to the specific context.

Clinic workflow and the time of surgeons and anesthesiologists who are trained to perform surgery and maintain patients optimally around the time of surgery can be optimized if the brief advice provided by the surgeon/anesthetist is followed by prompt referral to a smoking cessation expert, tobacco Quitline or smoking clinic integrated into the perioperative service. In a busy perioperative service in a Canadian hospital, this type of care integration increased the rate of identifying smokers; pre-implementation only 8.6% of patients were asked about smoking whereas, post-implementation, this rose to 86.0%. Consequently, the rate of providing an initial treatment intervention to smokers increased from 0% to 67.5% but there was no data on whether this improved smoking cessation rates [62].

There is a crucial need to establish smoking cessation training programs across the spectrum of perioperative care providers, including within residency curricula [58]. Increasing clinicians' knowledge leads to increases in smoking cessation delivery to patients [63]. A Cochrane review of 17 RCTs from mostly high-income countries demonstrated that training health professionals to provide smoking cessation interventions increased professional performance on asking, counselling, and assisting smokers in their quit attempt, and increased continuous abstinence from smoking (OR 1.60, 95% CI 1.26 to 2.03, p = 0.03) [64]. In the perioperative setting, technology can be leveraged to increase clinician engagement, by delivering training in formats (e.g., online) that will allow clinicians to earn CME credit and include pathways to reimbursement for example, the American College of Surgeons Division of Education Surgical Smoking Cessation in the Surgical Patient program 1-hour online course.

Misconceptions that patients are too nervous about surgery to discuss smoking cessation needs to be addressed through education. There is evidence that patients who smoke want to quit [7], and we found in this review that surgery itself is a facilitator for smoking cessation

[65,66]. Smoking cessation before surgery reduces postoperative complications [67–69] and practice guidelines recommend smoking cessation at least 4 weeks before surgery [13]. In clinical practice, patients are often seen just before surgery, and this may inform the belief that smoking cessation interventions are not worthwhile given the short time frame in which patients are seen. However, there is evidence that smoking cessation even a few days before surgery reduces the risk of post-surgical complications [70]. Moreover, beyond reducing complications, helping patients quit smoking for surgery has the potential for long-term abstinence [71] and therefore long-term health benefits.

We found common barriers to smoking cessation related to emotions, such as smoking for enjoyment or stress relief, particularly in cancer patients, and social influences, including healthcare providers' smoking behaviors. This suggests the necessity for tailored smoking cessation strategies in this group, emphasizing the improved prognosis for those who quit even after a cancer diagnosis [72]. Family involvement may be beneficial, and the influence of healthcare providers' attitudes on their smoking habits needs further study. A recent systematic review and meta-analysis of 246 studies and 497,081 physicians, found that although there was a downward trend in smoking prevalence among physicians, it was still high at 21% (95% CI 20% - 23%) [73]. None of our identified studies evaluated the impact of tobacco cessation among healthcare providers on smoking outcomes for patients. Hospital smoking cessation programs need to account for health care provider smoking behavior.

## Limitations

Limitations of this review include firstly, we only including articles published in English which were heterogenous and of varied sample sizes, potentially limiting its generalizability to other non-English speaking settings and with different sociocultural and environmental contexts. Secondly, we were unable to quantify whether the identified barriers and facilitators were considered trivial, moderate, or large as this was not assessed in the individual studies. Thirdly, our theme choices were guided by the TDF rather than by strictly inductive analysis. This might have led us to miss unique or unexpected themes that didn't fit the established TDF domains. Lastly, the subjective nature of our quality assessment with the MMAT scoring, might have introduced some author bias.

## Conclusions

Key domains that influence perioperative smoking cessation are environmental context and resources, knowledge, beliefs about capabilities, professional roles and identities and reinforcement. Specific barriers from providers include lack of time, lack of knowledge of smoking cessation interventions and smoking cessations support. Specific barriers from patients include failure of providers to advise them to quit whereas surgery itself serves as a motivator to quit smoking. High-quality evidence is needed to determine whether modifying these factors will impact patient smoking outcomes.

## Supporting information

**S1 Fig. Number of barriers and facilitators mapped to each TDF domain.**
(DOCX)

**S1 Table. Preferred Reporting Items for Systematic reviews and Meta-Analyses extension for Scoping Reviews (PRISMA-ScR) checklist.**
(DOCX)

**S2 Table. Search strategies.**
(DOCX)

**S3 Table. Quality assessment of included studies with the Mixed Methods Appraisal Tool (MMAT).**
(DOCX)

**S1 File. Reports excluded at full text screening stage.**
(DOCX)

## Author Contributions

**Conceptualization:** Sandra Ofori, Flavia K. Borges, Maura M. Marcucci, Lawrence Mbuagbaw, P. J. Devereaux.

**Data curation:** Daniel Rayner, David Mikhail.

**Formal analysis:** Sandra Ofori, Lawrence Mbuagbaw.

**Methodology:** Sandra Ofori, Flavia K. Borges, Maura M. Marcucci, David Conen, Lawrence Mbuagbaw, P. J. Devereaux.

**Writing – original draft:** Sandra Ofori.

**Writing – review & editing:** Sandra Ofori, Daniel Rayner, David Mikhail, Flavia K. Borges, Maura M. Marcucci, David Conen, Lawrence Mbuagbaw, P. J. Devereaux.

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
