## [Decision Letter · Decision Letter 0]

14 Jun 2023

PONE-D-23-16365Barriers and facilitators to preoperative smoking cessation: a scoping reviewPLOS ONE

Dear Dr. OFORI,

Thank you for submitting your manuscript to PLOS ONE. After careful consideration, we feel that it has merit but does not fully meet PLOS ONE’s publication criteria as it currently stands. Therefore, we invite you to submit a revised version of the manuscript that addresses the points raised during the review process.

ACADEMIC EDITOR:

please assess all the reviewers comments 

We look forward to receiving your revised manuscript.

Kind regards,

Silvia Fiorelli

Academic Editor

PLOS ONE

Journal Requirements:

5. We note that this manuscript is a systematic review or meta-analysis; our author guidelines therefore require that you use PRISMA guidance to help improve reporting quality of this type of study. Please upload copies of the completed PRISMA checklist as Supporting Information with a file name “PRISMA checklist”.

Reviewers' comments:

Reviewer's Responses to Questions

**Comments to the Author**

1. Is the manuscript technically sound, and do the data support the conclusions?

Reviewer #1: Yes

Reviewer #2: Yes

2. Has the statistical analysis been performed appropriately and rigorously? 

Reviewer #1: N/A

Reviewer #2: N/A

3. Have the authors made all data underlying the findings in their manuscript fully available?

Reviewer #1: Yes

Reviewer #2: Yes

4. Is the manuscript presented in an intelligible fashion and written in standard English?

Reviewer #1: Yes

Reviewer #2: Yes

5. Review Comments to the Author

Reviewer #1: State source of funding

The Short title applies to a wider scope

Please clarify what informed the choice of the search engines

Background- the figure 2.4% , not clear if it applies to the whole population or only to smokers

Provide key or footnote in your diagrams and tables

Reviewer #2: Thanks for submitting this manuscript.

I have multiple points to the authors to consider or to clarify on the manuscript:

1- Please state the qualifications of the reviewers including the third one. Line 122 and 126.

2- Could you state whether the selected studies were theory-base or not. Table 1.

3- Please review line 260 for miss spelling reported.

4- Please review line 382 for clarify.

5- I suggest to split table 2 to two tables. One for health care providers and one for patients only for easy reading.

Best wishes,

6. PLOS authors have the option to publish the peer review history of their article (what does this mean?). If published, this will include your full peer review and any attached files.

Reviewer #1: No

Reviewer #2: No

---

## [Author Response · Author response to Decision Letter 0]

29 Jun 2023

We have responded to all reviewers and editor comments in the "Response to Reviewers" document attached to this submission. Thank you.

---

## [Decision Letter · Decision Letter 1]

4 Sep 2023

PONE-D-23-16365R1Barriers and facilitators to perioperative smoking cessation: a scoping reviewPLOS ONE

Dear Dr. OFORI,

Thank you for submitting your manuscript to PLOS ONE. After careful consideration, we feel that it has merit but does not fully meet PLOS ONE’s publication criteria as it currently stands. Therefore, we invite you to submit a revised version of the manuscript that addresses the points raised during the review process.

ACADEMIC EDITOR:

please carefully assess all the reviewers comments

We look forward to receiving your revised manuscript.

Kind regards,

Silvia Fiorelli

Academic Editor

PLOS ONE

Journal Requirements:

Reviewers' comments:

Reviewer's Responses to Questions

**Comments to the Author**

1. If the authors have adequately addressed your comments raised in a previous round of review and you feel that this manuscript is now acceptable for publication, you may indicate that here to bypass the “Comments to the Author” section, enter your conflict of interest statement in the “Confidential to Editor” section, and submit your "Accept" recommendation.

Reviewer #2: All comments have been addressed

Reviewer #3: (No Response)

2. Is the manuscript technically sound, and do the data support the conclusions?

Reviewer #2: Yes

Reviewer #3: Partly

3. Has the statistical analysis been performed appropriately and rigorously? 

Reviewer #2: Yes

Reviewer #3: N/A

4. Have the authors made all data underlying the findings in their manuscript fully available?

Reviewer #2: Yes

Reviewer #3: No

5. Is the manuscript presented in an intelligible fashion and written in standard English?

Reviewer #2: Yes

Reviewer #3: Yes

6. Review Comments to the Author

Reviewer #2: (No Response)

Reviewer #3: Introduction, line 68 – I think you mean systems-led behavior change or system-level support to lead to behavior change?

Suggested copy edit:

The theoretical domains framework (TDF) is an evidence-based, comprehensive framework that can help understand the determinants of behavior change [24].

Section 2.1, add appendix reference next to mesh terms so people know where they can find it.

Section 2.2 should be cleaned up; you are referring to eligibility of all the studies but it is a bit confusing with the participants and semi-colon. Maybe start broad e.g., criteria as it related to all studies and then specify that studies could include your specified population. You also seem to be missing more specification for your exclusion criteria, a lot of the articles you exclude on in the PRISMA are not specified criteria in this section.

These sentences are found in the section above eligibility criteria, but are not in the appropriate section:

The search was restricted to 109 English language articles published in 2000 or later.

AND

We screened systematic reviews 111 for individual studies that met the eligibility criteria.

Could the authors include a citation for the approach to quality reporting (section 2.5) starting on line 138-9? Is this standard practice, some justification for this approach is necessary.

I find this sentence really confusing in the data synthesis section: From quantitative studies, we summarized the proportion of participants who report negative or permissive attitudes towards smoking and provide smoking cessation support/interventions as numbers and percentages. I don’t think a reader could replicate this, could you be clearer in how someone would reproduce this approach.

You use HCP on page 7 line 173 for the first time and don’t define it.

Results, the authors don’t really address that family members are included in some of these studies as are there perspectives, should this not be added in some way beyond Table 1, or if the authors choose not to report it, state that and why? It looks like it is only brought up once in the results.

It seems like the authors report on “other” in Table 2, is the family members? Could you better define this in the table – what is this subgroup comprised of?

Could the authors explain how they differentiate between capabilities vs. motivation? Are these articles actually talking about self-efficacy overall or is motivation salient here, if so, should the table be updated?

Aren’t these two items in table 2, category 6, not knowledge issues? Misperceptions?

- Barrier: indication for surgery not related to smoking

- Barrier: underestimation of the effect of smoking cessation on postop complications

Table 2, category 8, is lack of intentions the right word here? The way it os described in the table doesn’t seem like it’s a lack of intention on the part of the providers but a lack of belief in the outcome from the providers perspective, also by stage of change do you mean perceived readiness to change?

Also, a general comment is you do have a category that is decision processes, it seems to me that anything related to stages of change or motivation to change should go here - Suggest revising.

Could you walk me through why pessimism fits in the optimism them? It seems like the opposite.

The authors need to specifically cite the studies that this relates to in the following sentence on Page 15 - Lack of hospital support regarding accessibility and availability of quit kits [cite], Quitline referral materials [cite], nicotine replacement therapy [cite], and lack of integration of care between all professionals [cite] and hospital environments [cite] involved in the patient's care were some of the identified barriers within this theme [23].

Page 15, suggest you add what country this is in as there is a lot of variation in this study: In one study of 2,740 surgeons, 215 40.2% strongly agreed with the statement “reimbursement issues prevent me from providing 216 tobacco-use cessation services”?

Under the knowledge heading on page 16, the citations should not all be piled at the end but linked to the specific points in the sentence, please revise.

Under the knowledge heading, was this pattern globally providers had a misconception? Was it restricted to certain regions? This would be important specification to add.

Skills section, first sentence could you move the citations to the training or awareness section so the reader can know what article links to what statement?

Intentions heading, you have readiness to change in the table, but you don’t comment on it in this section.

For the key findings could you also numerically add the number of studies that outline these findings for each point? e.g. barriers (n) vs facilitators (n)

Line 372- discussion is an important point, however, given the breadth of studies this review covers, it is a little too broad of a comment. The authors should consider how this comment relates to different health care systems which articles in this review cover, I would imagine some countries do have very different reimbursements or even some specialities might have more integration than others? I would suggest the authors spend some time looking at the differences and revise this section accordingly, so it isn’t just western based recommendations or at least accounts for different healthcare systems.

Same reflection to line 391 – discussion, is the Cochrane review restricted to certain regions or healthcare systems? Specify

The first part of this sentence needs a citation: There is compelling evidence that patients who smoke want to quit, [cite] and we found in this review that surgery itself is a facilitator for smoking cessation [65,66].

Is this assertion true for this part of the sentence on line 411, discussion? - helping patients quit smoking for surgery has the potential for long-term abstinence? – could you cite

Limitations – given that you used a theoretical framework you tried to fit your themes into this framework rather than generate them inductively, you may want to mention this, I also think the authors needs to comment on generalizability as some of the articles were very small samples vs others and that there was large heterogeneity across the articles. Also missing authors bias, there seems to be a high level of subjectivity about the quality of the articles and the MMAT assessment, it seems like this should be acknowledged.

Figure 1 prisma, you have *’s that don’t link with anything, could you add footnotes for these symbols and make sure you include why records were excluded in the first part of the diagram too, not just the last.

Comment on supplemental materials – the formatting looks a bit messy for the search criteria - extra letters ect. like it was just thrown together, could you clean these up or make the tables consistent in some way?

Table 1 looks ok to me.

7. PLOS authors have the option to publish the peer review history of their article (what does this mean?). If published, this will include your full peer review and any attached files.

Reviewer #2: No

Reviewer #3: **Yes: **Teresa DeAtley

---

## [Author Response · Author response to Decision Letter 1]

6 Sep 2023

Please see submitted "Response to reviewers" document. Thank you.

---

## [Decision Letter · Decision Letter 2]

7 Nov 2023

PONE-D-23-16365R2Barriers and facilitators to perioperative smoking cessation: a scoping reviewPLOS ONE

Dear Dr. ONOFRI,

Thank you for submitting your manuscript to PLOS ONE. After careful consideration, we feel that it has merit but does not fully meet PLOS ONE’s publication criteria as it currently stands. Therefore, we invite you to submit a revised version of the manuscript that addresses the points raised during the review process.

ACADEMIC EDITOR: Pplease carefully assess all the reviewers comments 

We look forward to receiving your revised manuscript.

Kind regards,

Silvia Fiorelli

Academic Editor

PLOS ONE

Journal Requirements:

Reviewers' comments:

Reviewer's Responses to Questions

**Comments to the Author**

1. If the authors have adequately addressed your comments raised in a previous round of review and you feel that this manuscript is now acceptable for publication, you may indicate that here to bypass the “Comments to the Author” section, enter your conflict of interest statement in the “Confidential to Editor” section, and submit your "Accept" recommendation.

Reviewer #2: All comments have been addressed

Reviewer #4: (No Response)

2. Is the manuscript technically sound, and do the data support the conclusions?

Reviewer #2: Yes

Reviewer #4: Yes

3. Has the statistical analysis been performed appropriately and rigorously? 

Reviewer #2: I Don't Know

Reviewer #4: Yes

4. Have the authors made all data underlying the findings in their manuscript fully available?

Reviewer #2: Yes

Reviewer #4: Yes

5. Is the manuscript presented in an intelligible fashion and written in standard English?

Reviewer #2: Yes

Reviewer #4: Yes

6. Review Comments to the Author

Reviewer #2: (No Response)

Reviewer #4: 1. In general, the background has a bit too much detail on prior work (particularly lines 80-88) - would keep intro more succinct and add this level of detail to the discussion.

2. Commas are overused in several places in the manuscript

3. "Hand-searched" is an odd term - were the hands actually used to search? Or are you indicating that the references were individually reviewed by the authors?

4. It is generally frowned upon to refer to patients by only the noun "smokers" (line 114) instead of "patients who were active smokers" or "patients who smoke".

5. Its confusing to the readers how RCTs would address barriers/facilitators in the methods section, though clarified later when it is stated that they had additional qualitative analysis performed.

6. were the 10 pilot screening references chosen at random? Or were they the first (and likely most relevant) 10?

7. Please provide more detail about Covidence (company, location is usually appropriate when referring to a software in a scientific manuscript)

8. Do not repeat data that is in tables in text (including location of studies). This will help with the referenced word limits in response to other reviewer comments.

9. Table 2 - "physicians and surgeons should provide advice to quit smoking..." how is a "should" statement a facilitator? Do you mean that if they BOTH did it it was a facilitator? It's confusing.

10. How are perceived health benefits a barrier - does that mean "lack of" perceived benefits?

11. Knowledge "on" is used often - do they mean knowledge of?

12. Why is 8: lack of intention - "providers' belief.." not in the belief section?

13. In the discussion authors state (line 233) that there is a "misconception" that interventions were not efficacious - in fact recent systematic reviews have supported this conception.

14. Must of the discussion is biased in statements that interventions help but not supported by the data - for example line 301-302 reports on beliefs but lacks robust data as to actual outcomes.

15. Similarly in lines 388-390

16. Overall discussion rehashes much of the results without substantial additional information - would be helpfful to make more concise and summarize what of your results is new/additive to the literature in each area.

7. PLOS authors have the option to publish the peer review history of their article (what does this mean?). If published, this will include your full peer review and any attached files.

Reviewer #2: No

Reviewer #4: No

---

## [Author Response · Author response to Decision Letter 2]

23 Nov 2023

We have included a "Response to Reviewers" document wherein we have addressed all comments. Thank you.

---

## [Decision Letter · Decision Letter 3]

22 Jan 2024

Barriers and facilitators to perioperative smoking cessation: a scoping review

PONE-D-23-16365R3

Dear Dr. OFORI,

We’re pleased to inform you that your manuscript has been judged scientifically suitable for publication and will be formally accepted for publication once it meets all outstanding technical requirements.

Kind regards,

Silvia Fiorelli

Academic Editor

PLOS ONE

Additional Editor Comments (optional):

Congratulations to the authors and thanks to the reviewers for the provided suggestions which really helped improve the quality of the manuscript

Reviewers' comments:

Reviewer's Responses to Questions

**Comments to the Author**

1. If the authors have adequately addressed your comments raised in a previous round of review and you feel that this manuscript is now acceptable for publication, you may indicate that here to bypass the “Comments to the Author” section, enter your conflict of interest statement in the “Confidential to Editor” section, and submit your "Accept" recommendation.

Reviewer #4: All comments have been addressed

2. Is the manuscript technically sound, and do the data support the conclusions?

Reviewer #4: Yes

3. Has the statistical analysis been performed appropriately and rigorously? 

Reviewer #4: Yes

4. Have the authors made all data underlying the findings in their manuscript fully available?

Reviewer #4: Yes

5. Is the manuscript presented in an intelligible fashion and written in standard English?

Reviewer #4: Yes

6. Review Comments to the Author

Reviewer #4: Thank you for the detailed responses. Improved with appropriate edits.

Trying to meet minimum character count.

7. PLOS authors have the option to publish the peer review history of their article (what does this mean?). If published, this will include your full peer review and any attached files.

Reviewer #4: No

---

## [Editor Report · Acceptance letter]

21 Feb 2024

PONE-D-23-16365R3 

PLOS ONE

Dear Dr. OFORI, 

I'm pleased to inform you that your manuscript has been deemed suitable for publication in PLOS ONE. Congratulations! Your manuscript is now being handed over to our production team.

Kind regards, 

on behalf of

Dr. Silvia Fiorelli 

Academic Editor

PLOS ONE